# NOVEL VIEW SYNTHESIS WITH DIFFUSION MODELS

**Daniel Watson**
Google Research, Brain

**William Chan**
Google Research, Brain

**Ricardo Martin-Brualla**
Google Research

**Jonathan Ho**
Google Research, Brain

**Andrea Tagliasacchi**
Google Research, Brain

**Mohammad Norouzi**
Google Research, Brain

## ABSTRACT

We present 3DiM, a diffusion model for 3D novel view synthesis, which is able to translate a single input view into consistent and sharp completions across many views. The core component of 3DiM is a pose-conditional image-to-image diffusion model, which is trained to take a source view and its pose as inputs, and generates a novel view for a target pose as output. 3DiM can then generate multiple views that are approximately 3D consistent using a novel technique called *stochastic conditioning*. At inference time, the output views are generated autoregressively. When generating each novel view, one selects a random conditioning view from the set of previously generated views at each denoising step. We demonstrate that stochastic conditioning significantly improves 3D consistency compared to a naïve sampler for an image-to-image diffusion model, which involves conditioning on a single fixed view. We compare 3DiM to prior work on the SRN ShapeNet dataset, demonstrating that 3DiM's generated completions from a single view achieve much higher fidelity, while being approximately 3D consistent. We also introduce a new evaluation methodology, *3D consistency scoring*, to *quantify* the 3D consistency of a generated object by training a neural field on the model's output views. 3DiM is geometry free, does not rely on hyper-networks or test-time optimization for novel view synthesis, and allows a single model to easily scale to a large number of scenes.

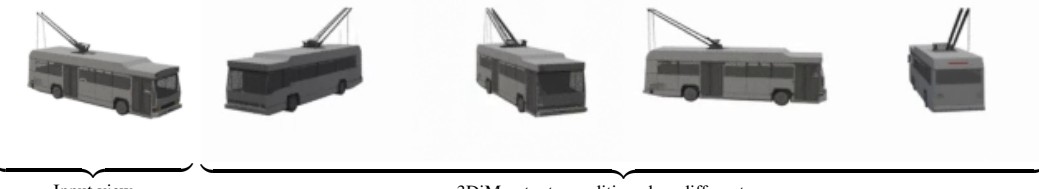

Input view      3DiM outputs conditioned on different poses

Figure 1: Given a single input image on the left, 3DiM performs novel view synthesis and generates the four views on the right. We trained a single ∼471M parameter 3DiM on *all* of ShapeNet (without class-conditioning) and sample frames with 256 steps (512 score function evaluations with classifier-free guidance). See the Supplementary Website (https://3d-diffusion.github.io/) for video outputs.

## 1 INTRODUCTION

Diffusion Probabilistic Models (DPMs) (Sohl-Dickstein et al., 2015; Song & Ermon, 2019; Ho et al., 2020), also known as simply *diffusion models*, have recently emerged as a powerful family of generative models, achieving state-of-the-art performance on audio and image synthesis (Chen et al., 2020; Dhariwal & Nichol, 2021), while admitting better training stability over adversarial approaches (Goodfellow et al., 2014), as well as likelihood computation, which enables further applications such as compression and density estimation (Song et al., 2021; Kingma et al., 2021). Diffusion models have achieved impressive empirical results in a variety of image-to-image translation tasks not limited to text-to-image, super-resolution, inpainting, colorization, uncropping, and artifact removal (Song et al., 2020; Saharia et al., 2021a; Ramesh et al., 2022; Saharia et al., 2022).

One particular image-to-image translation problem where diffusion models have not been investigated is *novel view synthesis*, where, given a set of images of a given 3D scene, the task is to infer how the scene looks from novel viewpoints. Before the recent emergence of Scene Representation Networks (SRN) (Sitzmann et al., 2019) and Neural Radiance Fields (NeRF) (Mildenhall et al., 2020), state-of-the-art approaches to novel view synthesis were typically built on generative models (Sun et al., 2018) or more classical techniques on interpolation or disparity estimation (Park et al., 2017; Zhou et al., 2018). Today, these models have been outperformed by NeRF-class models (Yu et al., 2021; Niemeyer et al., 2021; Jang & Agapito, 2021), where 3D consistency is *guaranteed by construction*, as images are generated by volume rendering of a single underlying 3D representation (a.k.a. "geometry-aware" models).

Still, these approaches feature different limitations. Heavily regularized NeRFs for novel view synthesis with few images such as RegNeRF (Niemeyer et al., 2021) produce undesired artifacts when given very few images, and fail to leverage knowledge from multiple scenes (recall NeRFs are trained on a single scene, i.e., one model *per scene*), and given one or very few views of a novel scene, a reasonable model must *extrapolate* to complete the occluded parts of the scene. Pixel-NeRF (Yu et al., 2021) and VisionNeRF (Lin et al., 2022) address this by training NeRF-like models conditioned on feature maps that encode the novel input view(s). However, these approaches are *regressive* rather than *generative*, and as a result, they cannot yield different plausible modes and are prone to blurriness. This type of failure has also been previously observed in regression-based models (Saharia et al., 2021b). Other works such as CodeNeRF (Jang & Agapito, 2021) and LoL-NeRF (Rebain et al., 2021) instead employ test-time optimization to handle novel scenes, but still have issues with sample quality.

In recent literature, *geometry-free* approaches (i.e., methods without explicit geometric inductive biases like those introduced by volume rendering) such as Light Field Networks (LFN) (Sitzmann et al., 2021) and Scene Representation Transformers (SRT) (Sajjadi et al., 2021) have achieved results competitive with 3D-aware methods in the "few-shot" setting, where the number of conditioning views is limited (i.e., 1-10 images vs. dozens of images as in the usual NeRF setting). Similarly to our approach, EG3D (Chan et al., 2022) provides *approximate* 3D consistency by leveraging generative models. EG3D employs a StyleGAN (Karras et al., 2019) with volumetric rendering, followed by generative super-resolution (the latter being responsible for the approximation). In comparison to this complex setup, we do not only provide a significantly simpler architecture, but also a simpler hyper-parameter tuning experience compared GANs, which are well-known to be notoriously difficult to tune (Mescheder et al., 2018).

Motivated by these observations and the success of diffusion models in image-to-image tasks, we introduce *3D Diffusion Models* (3DiMs). 3DiMs are image-to-image diffusion models trained on *pairs* of images of the same scene, where we assume the poses of the two images are known. Drawing inspiration from Scene Representation Transformers (Sajjadi et al., 2021), 3DiMs are trained to build a conditional generative model of one view given another view and their poses. Our key discovery is that we can turn this image-to-image model into a model that can produce an entire set of 3D-consistent frames through autoregressive generation, which we enable with our novel *stochastic conditioning* sampling algorithm. We cover stochastic conditioning in more detail in Section 2.2 and provide an illustration in Figure 3. Compared to prior work, 3DiMs are generative (vs. regressive) geometry free models, they allow training to scale to a large number of scenes, and offer a simple end-to-end approach.

We now summarize our core contributions:

1. We introduce 3DiM, a geometry-free image-to-image diffusion model for novel view synthesis.
2. We introduce the *stochastic conditioning* sampling algorithm, which encourages 3DiM to generate 3D-consistent outputs.
3. We introduce *X-UNet*, a new UNet architecture (Ronneberger et al., 2015) variant for 3D novel view synthesis, demonstrating that changes in architecture are critical for high fidelity results.
4. We introduce an evaluation scheme for geometry-free view synthesis models, *3D consistency scoring*, that can numerically capture 3D consistency by training neural fields on model outputs.

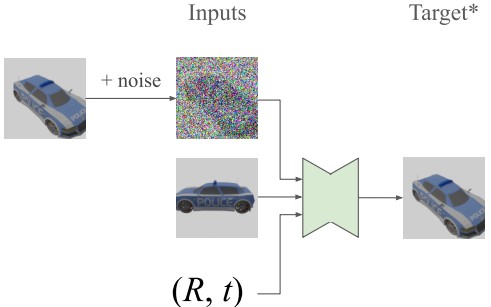

Figure 2: **Pose-conditional image-to-image training** – Example training inputs and outputs for pose-conditional image-to-image diffusion models, presented in Section 2.1. Given two frames from a common scene and their poses $(R, t)$, the training task is to undo the noise added to one of the two frames. (*) In practice, our neural network is trained to predict the Gaussian noise $\epsilon$ used to corrupt the original view – the predicted view is still just a linear combination of the noisy input and the predicted $\epsilon$.

## 2 POSE-CONDITIONAL DIFFUSION MODELS

To motivate 3DiMs, let us consider the problem of novel view synthesis given few images from a probabilistic perspective. Given a complete description of a 3D scene $\mathcal{S}$, for any pose $p$, the view $x^{(p)}$ at pose $p$ is fully determined from $\mathcal{S}$, i.e., views are conditionally independent given $\mathcal{S}$. However, we are interested in modeling distributions of the form $q(x_1, ..., x_m | x_{m+1}, ..., x_n)$ *without* $\mathcal{S}$, where views are no longer conditionally independent. A concrete example is the following: given the back of a person's head, there are multiple plausible views for the front. An image-to-image model sampling front views given only the back should indeed yield different outputs for each front view – with no guarantees that they will be consistent with each other – especially if it learns the data distribution perfectly. Similarly, given a single view of an object that appears small, there is ambiguity on the pose itself: is it small and close, or simply far away? Thus, given the inherent ambiguity in the few-shot setting, we need a sampling scheme where generated views can depend on each other in order to achieve 3D consistency. This contrasts NeRF approaches, where query *rays* are conditionally independent given a 3D representation $\mathcal{S}$ – an even stronger condition than imposing conditional independence among frames. Such approaches try to learn the richest possible representation for a single scene $\mathcal{S}$, while 3DiM avoids the difficulty of learning a generative model for $\mathcal{S}$ altogether.

### 2.1 IMAGE-TO-IMAGE DIFFUSION MODELS WITH POSE CONDITIONING

Given a data distribution $q(x_1, x_2)$ of pairs of views from a common scene at poses $p_1, p_2 \in \mathrm{SE}(3)$, we define an isotropic Gaussian process that adds increasing amounts of noise to data samples as the signal-to-noise-ratio $\lambda$ decreases, following Salimans & Ho (2022):

$$q(z_k^{(\lambda)} | x_k) := \mathcal{N}(z_k^{(\lambda)}; \sigma(\lambda)^{\frac{1}{2}} x_k, \sigma(-\lambda)\mathbf{I}) \tag{1}$$

where $\sigma(\cdot)$ is the sigmoid function. We can apply the reparametrization trick (Kingma & Welling, 2013) and sample from these marginal distributions via

$$z_k^{(\lambda)} = \sigma(\lambda)^{\frac{1}{2}} x_k + \sigma(-\lambda)^{\frac{1}{2}} \epsilon, \quad \epsilon \sim \mathcal{N}(\mathbf{0}, \mathbf{I}) \tag{2}$$

Then, given a pair of views, we learn to reverse this process in one of the two frames by minimizing the objective proposed by Ho et al. (2020), which has been shown to yield much better sample quality than maximizing the true evidence lower bound (ELBO):

$$L(\theta) = \mathbb{E}_{q(x_1, x_2)} \ \mathbb{E}_{\lambda, \epsilon} \ \| \epsilon_\theta(z_2^{(\lambda)}, x_1, \lambda, p_1, p_2) - \epsilon \|_2^2 \tag{3}$$

where $\epsilon_\theta$ is a neural network whose task is to denoise the frame $z_2^{(\lambda)}$ given a different (clean) frame $x_1$, and $\lambda$ is the log signal-to-noise-ratio. To make our notation more legible, we slightly abuse notation and from now on we will simply write $\epsilon_\theta(z_2^{(\lambda)}, x_1)$. We illustrate training in Figure 2.

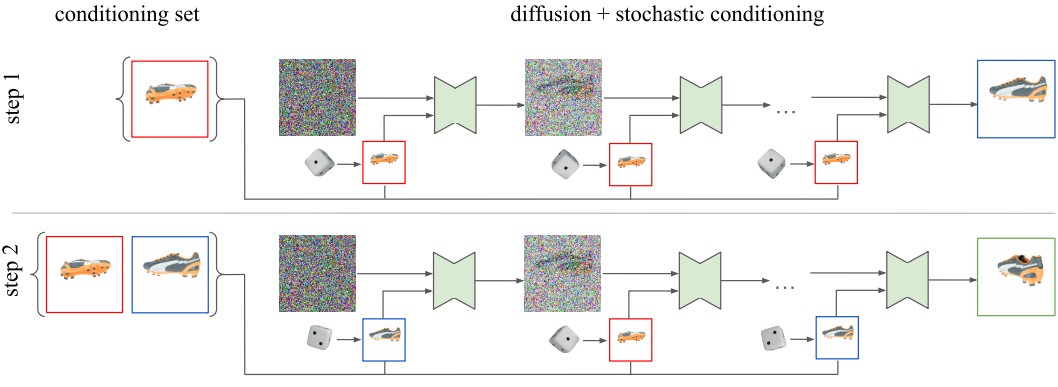

Figure 3: **Stochastic conditioning sampler** – We illustrate our proposed inference procedure for 3DiM, outlined in Section 2.2. There are two main components to our sampling procedure: (1) the autoregressive generation of multiple frames (illustrated vertically as "step 1", "step 2", etc.), and (2) the denoising process to generate each individual frame (illustrated horizontally). When generating a new frame, we select a previous frame as the conditioning frame *randomly* at each denoising step (illustrated with the dice). Note that this is not part of 3DiM training; also, we omit the pose inputs in the diagram to avoid overloading the figure.

## 2.2 3D CONSISTENCY VIA STOCHASTIC CONDITIONING

**Motivation.** We begin this section by motivating the need of our stochastic conditioning sampler. In the ideal situation, we would model our 3D scene frames using the chain rule decomposition:

$$p(\boldsymbol{x}) = \prod_i p(\boldsymbol{x}_i | \boldsymbol{x}_{<i}) \tag{4}$$

This factorization is ideal, as it models the distribution exactly without making any conditional independence assumptions. Each frame is generated autoregressively, conditioned on all the previous frames. However, we found this solution to perform poorly. Due to memory limitations, we can only condition on a limited number of frames in practice, (i.e., a $k$-Markovian model). We also find that, as we increase the maximum number of input frames $k$, the worse the sample quality becomes. In order to achieve the best possible sample quality, we thus opt for the bare minimum of $k = 2$ (i.e., an image-to-image model). Our key discovery is that, with $k = 2$, we can still achieve approximate 3D consistency. Instead of using a sampler that is Markovian over frames, we leverage the iterative nature of diffusion sampling by *varying* the conditioning frame at each denoising step.

**Stochastic Conditioning.** We now detail our novel *stochastic conditioning* sampling procedure that allows us to generate 3D-consistent samples from a 3DiM. We start with a set of conditioning views $\mathcal{X} = \{\boldsymbol{x}_1, ..., \boldsymbol{x}_k\}$ of a static scene, where typically $k = 1$ or is very small. We then generate a new frame by running a modified version of the standard denoising diffusion reverse process for steps $\lambda_{\min} = \lambda_T < \lambda_{T-1} < ... < \lambda_0 = \lambda_{\max}$:

$$\hat{\boldsymbol{x}}_{k+1} = \frac{1}{\sigma(\lambda_t)^{\frac{1}{2}}} \left( \boldsymbol{z}_{k+1}^{(\lambda_t)} - \sigma(-\lambda_t)^{\frac{1}{2}} \boldsymbol{\epsilon}_\theta(\boldsymbol{z}_{k+1}^{(\lambda_t)}, \boldsymbol{x}_i) \right) \tag{5}$$

$$\boldsymbol{z}_{k+1}^{(\lambda_{t-1})} \sim q\left( \boldsymbol{z}_{k+1}^{(\lambda_{t-1})} | \boldsymbol{z}_{k+1}^{(\lambda_t)}, \hat{\boldsymbol{x}}_{k+1} \right) \tag{6}$$

where, crucially, $i \sim \text{Uniform}(\{1, ..., k\})$ is re-sampled at each denoising step. In other words, each individual denoising step is conditioned on a different *random* view from $\mathcal{X}$ (the set that contains the input view(s) and the previously generated samples). Once we finish running this sampling chain and produce a final $\boldsymbol{x}_{k+1}$, we simply add it to $\mathcal{X}$ and repeat this procedure if we want to sample more frames. Given sufficient denoising steps, stochastic conditioning allows each generated frame to be guided by all previous frames. See Figure 3 for an illustration. In practice, we use 256 denoising steps, which we find to be sufficient to achieve both high sample quality and approximate 3D consistency. As usual in the literature, the first (noisiest sample) is just a Gaussian, i.e., $\boldsymbol{z}_i^{(\lambda_T)} \sim \mathcal{N}(\mathbf{0}, \mathbf{I})$, and at the last step $\lambda_0$, we sample noiselessly.

We can interpret stochastic conditioning as a naïve approximation to true autoregressive sampling that works well in practice. True autoregressive sampling would require a score model of the form

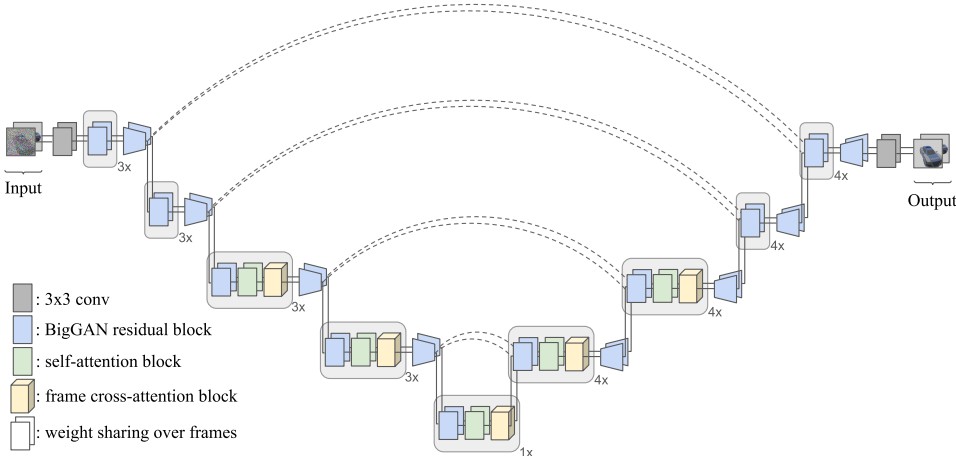

Figure 4: **X-UNet Architecture** – We modify the typical UNet architecture used by recent work on diffusion models to accomodate 3D novel view synthesis. We share the same UNet weights among the two input frames, the clean conditioning view and the denoising target view. We add cross attention layers to mix information between the input and output view, illustrated in yellow.

$\nabla_{\boldsymbol{z}_{k+1}^{(\lambda)}} \log q(\boldsymbol{z}_{k+1}^{(\lambda)} | \boldsymbol{x}_1, ..., \boldsymbol{x}_k)$, but this would strictly require multi-view training data, while we are ultimately interested in enabling novel view synthesis with as few as two training views per scene.

## 2.3  X-UNET

The 3DiM model needs a neural network architecture that takes both the conditioning frame and the noisy frame as inputs. One natural way to do this is simply to *concatenate* the two images along the channel dimensions, and use the standard UNet architecture (Ronneberger et al., 2015; Ho et al., 2020). This "Concat-UNet" has found significant success in prior work of image-to-image diffusion models (Saharia et al., 2021b;a). However, in our early experiments, we found that the Concat-UNet yields very poor results – there were severe 3D inconsistencies and lack of alignment to the conditioning image. We hypothesize that, given limited model capacity and training data, it is difficult to learn complex, nonlinear image transformations that only rely on self-attention. We thus introduce our *X-UNet*, whose core changes are (1) sharing parameters to process each of the two views, and (2) using cross attention between the two views. We find our X-UNet architecture to be very effective for 3D novel view synthesis.

We now describe X-UNet in detail. We follow Ho et al. (2020); Song et al. (2020), and use the UNet (Ronneberger et al., 2015) with residual blocks and self-attention.We also take inspiration from Video Diffusion Models (Ho et al., 2022) by sharing weights over the two input frames for all the convolutional and self-attention layers, but with several key differences:

1. We let each frame have its own noise level (recall that the inputs to a DDPM residual block are feature maps as well as a positional encoding for the noise level). We use a positional encoding of $\lambda_{\max}$ for the clean frame. Ho et al. (2022) conversely denoise multiple frames simultaneously, each at the same noise level.
2. Alike Ho et al. (2020), we modulate each UNet block via FiLM (Dumoulin et al., 2018), but we use the sum of pose and noise-level positional encodings, as opposed to the noise-level embedding alone. Our pose encoding additionally differs in that they are of the same dimensionality as frames– they are camera rays, identical to those used by Sajjadi et al. (2021).
3. Instead of attending over "time" after each self-attention layer like Ho et al. (2022), which in our case would entail only two attention weights, we define a *cross-attention* layer and let each frame's feature maps call this layer to query the other frame's feature maps.

For more details on our proposed architecture, we refer the reader to the Supplementary Material (Sec.6). We also provide a comparison to the "Concat-UNet" architecture in Section 3.2.

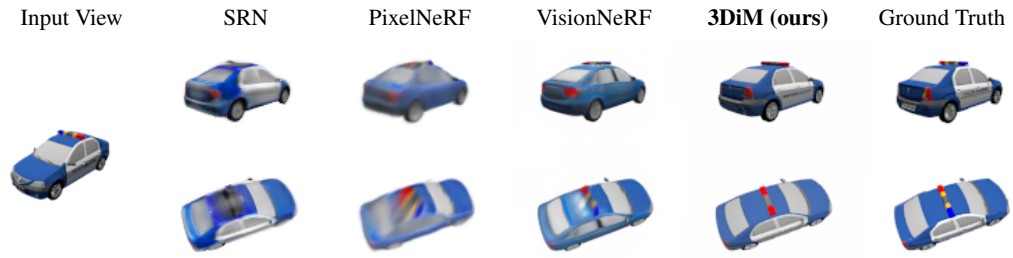

Figure 5: **State-of-the-art comparisons** – Example input and output views of a 3DiM trained on the SRN cars dataset at the 128x128 resolution, compared to existing geometry-aware methods. Results are best appreciated in the Supplementary Website (https://3d-diffusion.github.io/).

| | SRN cars | | | SRN chairs | | |
|---|---|---|---|---|---|---|
| | PSNR (↑) | SSIM (↑) | FID (↓) | PSNR (↑) | SSIM (↑) | FID (↓) |
| Geometry-aware | | | | | | |
| SRN | 22.25 | 0.88 | 41.21 | 22.89 | 0.89 | 26.51 |
| PixelNeRF | 23.17 | 0.89 | 59.24 | 23.72 | 0.90 | 38.49 |
| VisionNeRF | 22.88 | 0.90 | 21.31 | **24.48** | **0.92** | 10.05 |
| CodeNeRF | **23.80** | *0.91 | – | 23.66 | *0.90 | – |
| Geometry-free | | | | | | |
| LFN | 22.42 | *0.89 | – | 22.26 | *0.90 | – |
| ENR | 22.26 | – | – | 22.83 | – | – |
| **3DiM (ours)** | 21.01 | 0.57 | **8.99** | 17.05 | 0.53 | **6.57** |

Table 2: **State-of-the-art comparisons** – Results on the SRN ShapeNet benchmark comparing 3DiMs to prior work on novel view synthesis from a single image. (*) SSIM scores may not necessarily be computed with a Gaussian kernel following Wang et al. (2004) – in practice we observe this can lead to differences of up to 0.02. We report these marked numbers directly from prior work.

## 3 EXPERIMENTS

We benchmark 3DiMs on the SRN ShapeNet dataset (Sitzmann et al., 2019) to allow comparisons with prior work on novel view synthesis from a single image. This dataset consists of views and poses of car and chair ShapeNet (Chang et al., 2015) assets, rendered at the 128x128 resolution.

We compare 3DiMs with Light Field Networks (LFN) (Sitzmann et al., 2021) and Equivariant Neural Rendering (ENR) (Dupont et al., 2020), two competitive geometry-free approaches, as well as geometry-aware approaches including Scene Representation Networks (SRN) (Sitzmann et al., 2019), PixelNeRF (Yu et al., 2021), CodeNeRF (Jang & Agapito, 2021) and the recent VisionNeRF (Lin et al., 2022). We include standard metrics in the literature: peak signal-to-noise-ratio (PSNR), structural similarity (SSIM) (Wang et al., 2004), and the Fréchet Inception Distance (FID) (Heusel et al., 2017). To remain consistent with prior work, we evaluate over all scenes in the held-out dataset (each has 251 views), conditioning our models on a single view (view #64) to produce all other 250 views via a single call to our sampler outlined in Section 2.2, and then report the average PSNR and SSIM. Importantly, FID scores were computed with this same number of views, comparing the set of all generated views against the set of all the ground truth views. Because prior work did not include FID scores, we acquire the evaluation image outputs of SRN, PixelNeRF and VisionNeRF and compute the scores ourselves. We reproduce PSNR scores, and carefully note that some of the models do not follow Wang et al. (2004) on their SSIM computation (they use a uniform rather than Gaussian kernel), so we also recompute SSIM scores following (Wang et al., 2004). For more details, including hyperparameter choices, see Supplementary Material (Sec.7).

**Out-of-distribution poses on SRN chairs**. We note that the test split of the SRN ShapeNet chairs was unintentionally released with out-of-distribution poses compared to the train split: most test

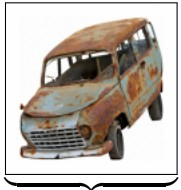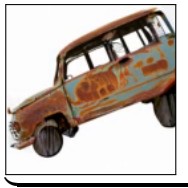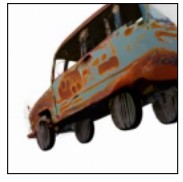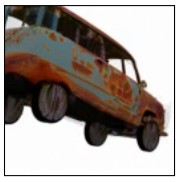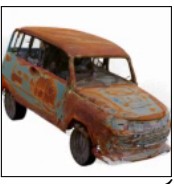

OOD input view          3DiM outputs conditioned on different *relative* poses

Figure 6: **3DiM samples on OOD synthetic images** – We qualitatively demonstrate the text-to-3D capabilities of 3DiM by synthethizing an input view with Imagen, a text-to-image model, and then create more views with 3DiM.

views are at a much larger distance to the object than those in the training dataset; we confirmed this via correspondence with the authors of Sitzmann et al. (2019). Because 3DiMs are geometry-free, we (unsurprisingly) observe that they do not perform well on this out-of-distribution evaluation task, as all the poses used at test-time are of scale never seen during training. However, simply merging, shuffling, and re-splitting the dataset completely fixes the issue. To maintain comparability with prior work, results on SRN chairs in Table 2 are those on the *original* dataset, and we denote the re-split dataset as "SRN chairs*" (note the *) in all subsequent tables to avoid confusion.

**State-of-the-art comparisons**. While 3DiM does not necessarily achieve superior reconstruction errors (PSNR and SSIM, see Table 2), qualitatively, we find that the fidelity of our generated videos can be strikingly better. We include an example in Figure 5. The use of diffusion models allows us to produce sharp samples, as opposed to regression models that are well-known to be prone to blurriness (Saharia et al., 2021b) in spite of high PSNR and SSIM scores. This is why we introduce and evaluate FID scores. In fact, we do not expect to achieve very low reconstruction errors due to the inherent ambiguity of novel view synthesis with a single image: constructing a consistent object with the given frame(s) is acceptable, but will still be punished by said reconstruction metrics if it differs from the ground truth views (even if achieving consistency). We additionally refer the reader to Section 3.1, where we include simple ablations demonstrating how worse models can improve the different standardized metrics in the literature. E.g., we show that with regression models, similarly to the baseline samples, PSNR and SSIM do not capture sharp modes well – they scores these models as much better despite the samples looking significantly more blurry (with PixelNeRF, which qualitatively seems the blurriest, achieving the best scores).

**Results on out-of-distribution images.**. We additionally show qualitative results attempting to lift single images in the wild (without known poses) into 3D. To this end, we train a 471M parameter 3DiM on *all* of ShapeNet (except for 10 objects per class which we leave out for testing). This model has identical parameters to the 3DiM used for SRN cars. However, it differs in that we poses with **relative** camera extrinsics (i.e., the camera position and rotation corresponding to the noisy input frame is always the same) as we do not know the poses of images in the wild. Our qualitative results on single-image-to-3D generation include:

1. ShapeNet objects held out from the training data
2. Images in the wild we directly took from the internet (with a white background, minimal shadow, and corresponding to any ShapeNet class)
3. Images synthesized by Imagen (Saharia et al., 2022), a text-to-image diffusion model

In order to make Imagen synthesize objects with white backgrounds and minimal shadows at the 128x128 resolution, in addition to natural language prompting, we inpaint a 5px white border at each denoising step when generating a 64x64 image, and then upsample it using the second model on the Imagen cascade, i.e., a text-conditional super-resolution model. We find that this strategy is much more reliable than only relying on prompt engineering to create images with white backgrounds: this leads the text-to-image model to make the rest of background consistent with the fully white border, so the generated image results in an out-of-distribution object while maintaining an in-distribution background. We present example novel views given a ShapeNet test object in Figure 1, and given a synthetic image from Imagen in Figure 6. We include several video samples for all the cases described above in the Supplementary Website (https://3d-diffusion.github.io/).

| | SRN cars | | | SRN chairs* | | |
| --- | --- | --- | --- | --- | --- | --- |
| | PSNR (↑) | SSIM (↑) | FID (↓) | PSNR (↑) | SSIM (↑) | FID (↓) |
| 3DiM | 21.01 | 0.57 | 8.99 | 14.29 | 0.48 | 3.30 |
| + no stochastic conditioning | **23.82** | 0.59 | **1.87** | 15.73 | 0.51 | **1.05** |
| + regression | 22.55 | **0.85** | 17.45 | 1**6.85** | **0.76** | 20.70 |

Table 3: **Ablation** – study removing stochastic conditioning and diffusion altogether from 3DiM. Results are included for novel view synthesis from a single image on the SRN ShapeNet benchmark. (*) Re-split chairs dataset as detailed in Section 3 – not comparable to numbers in Table 2.

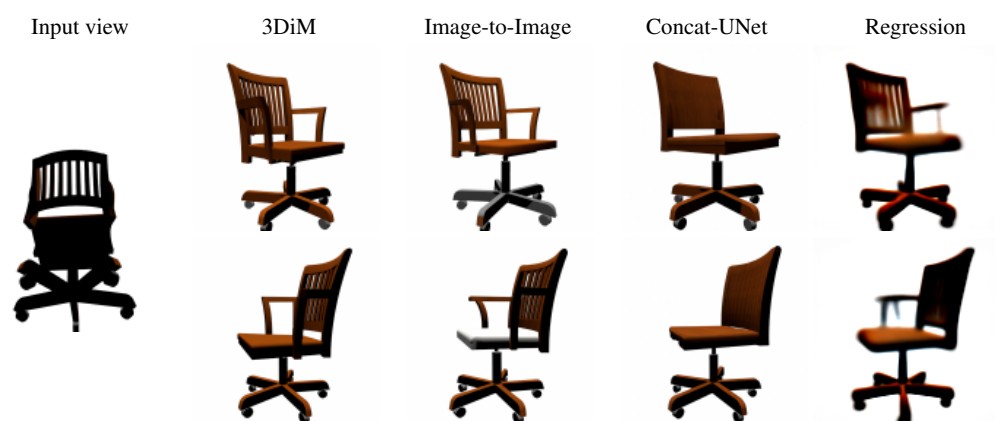

Figure 7: **Ablations** – Example input and output views of 3DiM ablations on the SRN chairs dataset at the 128x128 resolution. The second "Image-to-Image" column corresponds to removing our stochastic conditioning sampler from Section 2.2. The third "Concat-UNet" column corresponds to removing our proposed architecture in Section 2.3, resorting to a UNet that concatenates both images along the channel axis instead of weight sharing over frames. The fourth "Regression" column is one-step diffusion model trained from scratch.

## 3.1 ABLATION STUDIES

We now present some ablation studies on 3DiM. First, we remove our proposed sampler and use a naïve image-to-image model as discussed at the start of Section 2.2. We additionally remove the use of a diffusion process altogether, i.e., a regression model that generates samples via a single denoising step. Naturally, the regression models are trained separately, but we can still use an identical architecture, always feeding white noise for one frame, along with its corresponding noise level $\lambda_{min}$. Results are included in Table 3 and samples are included in Figure 7.

Unsurprisingly, we find that both of these components are crucial to achieve good results. The use of many diffusion steps allows sampling sharp images that achieve much better (lower) FID scores than both prior work and the regression models, where in both cases, reconstructions appear blurry. Notably, the regression models achieve better PSNR and SSIM scores than 3DiM despite the severe blurriness, suggesting that these standardized metrics fail to meaningfully capture sample quality for geometry-free models, at least when comparing them to geometry-aware or non-stochastic reconstruction approaches. Similarly, FID also has a failure mode: naïve image-to-image sampling improves (decreases) FID scores significantly, but severely worsens shape and texture inconsistencies between sampled frames. These findings suggest that none of these standardized metrics are sufficient to effectively evaluate geometry-free models for view synthesis; nevertheless, we observe that said metrics do correlate well with sample quality *across 3DiMs* and can still be useful indicators for hyperparameter tuning despite their individual failures.

| | cars | | | chairs* | | |
|---|---|---|---|---|---|---|
| | PSNR ($\uparrow$) | SSIM ($\uparrow$) | FID ($\downarrow$) | PSNR ($\uparrow$) | SSIM ($\uparrow$) | FID ($\downarrow$) |
| Concat-UNet | 17.21 | 0.52 | 21.54 | 12.36 | 0.44 | 5.15 |
| X-UNet | **21.01** | **0.57** | **8.99** | **14.29** | **0.48** | **3.30** |

Table 5: **Architecture comparison** – study comparing the X-UNet and Concat-UNet neural architectures. Results are included for novel view synthesis from a single image on the SRN ShapeNet benchmark, with the re-split chairs as detailed in Section 3. (*) Re-split chairs dataset as detailed in Section 3 – these numbers are not comparable to those in Table 2.

## 3.2 UNet Architecture Comparisons

In order to demonstrate the benefit of our proposed modifications, we additionally compare our proposed X-UNet architecture from Section 2.3 to the simpler UNet architecture following Saharia et al. (2021b;a) (which we simply call "Concat-UNet"). The Concat-UNet architecture, unlike ours, does not share weights across frames; instead, it simply concatenates the conditioning image to the noisy input image along the channel axis. To do this comparison, we train 3DiMs on the Concat-UNet architecture with the same number of hidden channels, and keep all other hyperparameters identical. Because our architecture has the additional cross-attention layer at the coarse-resolution blocks, our architecture has a slightly larger number of parameters than the Concat-UNet ($\sim$471M v.s. $\sim$421M). Results are included in Table 5 and Figure 7.

While the Concat-UNet architecture is able to sample frames that resemble the data distribution, we find that 3DiMs trained with our proposed X-UNet architecture suffer much less from 3D inconsistency and alignment to the conditioning frame. Moreover, while the metrics should be taken with a grain of salt as previously discussed, we find that *all* the metrics significantly worsen with the Concat-UNet. We hypothesize that our X-UNet architecture better exploits symmetries between frames and poses due to our proposed weight-sharing mechanism, and that the cross-attention helps significantly to align with the content of the conditioning view.

## 4 Evaluating 3D consistency in geometry-free view synthesis

As we demonstrate in Sections 3.1 and 3.2, the standardized metrics in the literature have failure modes when specifically applied to geometry-free novel view synthesis models, e.g., their inability to successfully measure 3D consistency, and the possibility of improving them with worse models. Leveraging the fact that volumetric rendering of colored density fields are 3D-consistent by design, we thus propose an additional evaluation scheme called "3D consistency scoring". Our metrics should satisfy the following desiderata:

1. The metric must penalize outputs that are not 3D consistent.
2. The metric must **not** penalize outputs that are 3D consistent but deviate from the ground truth.
3. The metric must penalize outputs that do not align with the conditioning view(s).

In order to satisfy the second requirement, we *cannot* compare output renders to ground-truth views. Thus, one straightforward way to satisfy all desiderata is to sample many views from the geometry-free model given a single view, train a NeRF-like neural field (Mildenhall et al., 2020) on a fraction of these views, and compute a set of metrics that compare neural field renders on the remaining views. This way, if the geometry-free model outputs inconsistent views, the training of neural field will be hindered and classical image evaluation metrics should clearly reflect this. Additionally, to enforce the third requirement, we simply include the conditioning view(s) that were used to generate the rest as part of the training data. We report PSNR, SSIM and FID on the held-out views, although one could use other metrics under our proposed evaluation scheme.

We evaluate 3DiMs on the SRN benchmark, and for comparison, we additionally include metrics for models trained on (1) the real test views and (2) on image-to-image samples from the 3DiMs, i.e., without our proposed sampler like we reported in Section 3. To maintain comparability, we sample the same number of views from the different 3DiMs we evaluate in this section, all at the same poses and conditioned on the same single views. We leave out 10% of the test views (25 out of 251) from neural field training, picking 25 random indices once and maintaining this choice of indices for all

| Training view source | SRN cars | | | SRN chairs* | | |
|---|---|---|---|---|---|---|
| | PSNR ($\uparrow$) | SSIM ($\uparrow$) | FID ($\downarrow$) | PSNR ($\uparrow$) | SSIM ($\uparrow$) | FID ($\downarrow$) |
| Original data (3D consistent) | 28.21 | 0.96 | 10.57 | 24.87 | 0.93 | 17.05 |
| 3DiM ($\sim$1.3B params) | 28.48 | **0.96** | 29.55 | **22.90** | **0.86** | **58.61** |
| 3DiM ($\sim$471M params) | **28.53** | **0.96** | **22.09** | 18.84 | 0.79 | 98.78 |
| + no stochastic conditioning | 25.78 | 0.94 | 30.51 | 17.61 | 0.75 | 116.16 |

Table 6: **3D consistency scores** – For neural fields trained with different view sources, we compare renders to held-out views from the same sources.

subsequent evaluations. We also train models with more parameters ($\sim$1.3B) in order to investigate whether increasing model capacity can further improve 3D consistency. See Supplementary Material (Sec.7) for more details on the neural fields we chose and their hyperparameters.

We find that our proposed evaluation scheme clearly punishes 3D inconsistency as desired – the neural fields trained on image-to-image 3DiM samples have worse scores across *all* metrics compared to neural fields trained on 3DiM outputs sampled via our stochastic conditioning. This helps quantify the value of our proposed sampler, and also prevents the metrics from punishing reasonable outputs that are coherent with the input view(s) but do not match the target views – a desirable property due to the stochasticity of generative models. Moreover, we qualitatively find that the smaller model reported in the rest of the paper is comparable in quality to the $\sim$1.3B parameter model on cars, though on chairs, we do observe a significant improvement on 3D consistency in our samples. Importantly, 3D consistency scoring agrees with our qualitative observations.

## 5 CONCLUSION AND FUTURE WORK

We propose 3DiM, a diffusion model for 3D novel view synthesis. Combining improvements in our X-UNet neural architecture (Section 2.3), with our novel *stochastic conditioning* sampling strategy that enables autoregressive generation over frames (Section 2.2), we show that from as few as a single image we can generate approximately 3D consistent views with very sharp sample quality. We additionally introduce *3D consistency scoring* to evaluate the 3D consistency of geometry-free generative models by training neural fields on model output views, as their performance will be hindered increasingly with inconsistent training data (Section 4). We thus show, both quantitatively and visually, that 3DiMs with stochastic conditioning can achieve approximate 3D consistency and high sample quality simultaneously, and how classical metrics fail to capture both sharp modes and 3D inconsistency.

A noteworthy limitation of 3DiM is that, due to the geometry-free setup, it can only handle distributions of poses that it is exposed to durining training. In particular, further study is required where more variations to the poses are introduced. For example, varying focal lengths, sensor widths, making the camera not look exactly at the objects, and varying the distances from the camera to the objects more aggressively. Our use of stochastic conditioning might also exacerbate the need for many denoising steps. While we believe that finding effective architectures that allow conditioning on sets of images remains an important open problem, stochastic conditioning unlocks the ability to train on examples that only have two views and helps overcome memory-intensive neural architectures. Similar approaches using randomized, sparse conditioning are already being explored in follow-up work on video generation (Davtyan et al., 2022) in order to enable efficient training when modeling the joint distribution is computationally prohibitive.

We are most excited about the possibility of applying 3DiM, which can model entire datasets with a single model, to the largest 3D datasets from the real world – though more research is required to handle noisy poses (due to the need for pose estimation) and other challenges present in this context. Developing end-to-end approaches for high-quality generation that are 3D consistent by design like more recent work (Müller et al., 2022) and that also yield high-quality 3D meshes remain important research problems. Another potentially significant application of 3DiM is its use as a prior in order to achieve 3D consistent generation via approaches that operate exclusively at sampling time (Poole et al., 2022; Zhou & Tulsiani, 2022; Xu et al., 2022) in order to enable wide adoption of similar, disruptive technologies on the 3D design industry.

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

# Novel View Synthesis with Diffusion Models

## Supplementary Material

## 6   ARCHITECTURE DETAILS

In order to maximize the reproducibility of our results, we provide code in JAX (Bradbury et al., 2018) for our proposed X-UNet neural architecture from Section 2.3. Assuming we have an input batch with elements each containing

```
{
    "z",  # noisy input image (HxWx3 tensor)
    "x",  # conditioning view (HxWx3 tensor)
    "logsnr",  # log signal-to-noise ratio of noisy image (scalar)
    "t",  # camera positions (two 3d vectors)
    "R",  # camera rotations (two 3x3 matrices)
    "K",  # camera intrinsics (a.k.a. calibration matrix) (3x3 matrix)
}
```

our neural network module is as follows:

```python
from typing import Optional

import flax.linen as nn
import jax.numpy as jnp
import numpy as onp
import visu3d as v3d

nonlinearity = nn.swish

def out_init_scale():
  """Zeros initializer."""
  return nn.initializers.variance_scaling(0.0, 'fan_in', 'truncated_normal')

def nearest_neighbor_upsample(h: jnp.ndarray):
  """Nearest neighbor upsampling for multiple frames."""
  B, F, H, W, C = h.shape
  h = h.reshape(B, F, H, 1, W, 1, C)
  h = jnp.broadcast_to(h, (B, F, H, 2, W, 2, C))
  return h.reshape(B, F, H * 2, W * 2, C)

def avgpool_downsample(h: jnp.ndarray, k: int = 2):
  """Average pooling downsampling for multiple frames."""
  return nn.avg_pool(h, (1, k, k), (1, k, k))

def posenc_ddpm(timesteps: jnp.ndarray, emb_ch: int, max_time: int = 1000.0):
  """Positional encodings for noise levels, following DDPM."""
  # 1000 is the magic number from DDPM. With different timesteps, we
  # normalize by the number of steps but still multiply by 1000.
  timesteps *= 1000.0 / max_time
  half_dim = emb_ch // 2
  # 10000 is the magic number from transformers.
  emb = onp.log(10000) / (half_dim - 1)
```

```python
    emb = jnp.exp(jnp.arange(half_dim, dtype=timesteps.dtype) * -emb)
    emb = emb.reshape(*([1] * (timesteps.ndim - 1)), emb.shape[-1])
    emb = timesteps[..., None] * emb
    emb = jnp.concatenate([jnp.sin(emb), jnp.cos(emb)], axis=-1)
    return emb

def posenc_nerf(x: jnp.ndarray, min_deg: int = 0, max_deg: int = 15):
    """Concatenate x and its positional encodings, following NeRF."""
    if min_deg == max_deg:
        return x
    scales = jnp.array([2**i for i in range(min_deg, max_deg)])
    xb = jnp.reshape(
        (x[..., None, :] * scales[:, None]), list(x.shape[:-1]) + [-1]
    )
    emb = jnp.sin(jnp.concatenate([xb, xb + onp.pi / 2.0], axis=-1))
    return jnp.concatenate([x, emb], axis=-1)

class GroupNorm(nn.Module):
    """Group normalization, applied over frames."""

    @nn.compact
    def __call__(self, h: jnp.ndarray):
        B, _, H, W, C = h.shape
        h = nn.GroupNorm(num_groups=32)(h.reshape(B * 2, 2, H, W, C))
        return h.reshape(B, 2, H, W, C)

class FiLM(nn.Module):
    """Feature-wise linear modulation."""

    features: int

    @nn.compact
    def __call__(self, h: jnp.ndarray, emb: jnp.ndarray):
        emb = nn.Dense(2 * self.features)(nonlinearity(emb))
        scale, shift = jnp.split(emb, 2, axis=-1)
        return h * (1.0 + scale) + shift

class ResnetBlock(nn.Module):
    """BigGAN-style residual block, applied over frames."""

    features: Optional[int] = None
    dropout: float = 0.0
    resample: Optional[str] = None

    @nn.compact
    def __call__(self, h_in: jnp.ndarray, emb: jnp.ndarray, *, train: bool):
        _, _, _, _, C = h_in.shape
        features = C if self.features is None else self.features

        h = nonlinearity(GroupNorm()(h_in))
        if self.resample is not None:
            updown = {
                'up': nearest_neighbor_upsample,
                'down': avgpool_downsample,
            }[self.resample]
```

```python
    h = updown(h)
    h_in = updown(h_in)

  h = nn.Conv(features, kernel_size=(1, 3, 3), strides=(1, 1, 1))(h)
  h = FiLM(features=features)(GroupNorm()(h), emb)
  h = nonlinearity(h)
  h = nn.Dropout(rate=self.dropout)(h, deterministic=not train)
  h = nn.Conv(
      features,
      kernel_size=(1, 3, 3),
      strides=(1, 1, 1),
      kernel_init=out_init_scale(),
  )(h)

  if C != features:
    h_in = nn.Dense(features)(h_in)
  return (h + h_in) / onp.sqrt(2)

class AttnLayer(nn.Module):
  """Attention layer usable for self and cross attention."""

  attn_heads: int = 4

  @nn.compact
  def __call__(self, *, q: jnp.ndarray, kv: jnp.ndarray):
    C = q.shape[-1]
    head_dim = C // self.attn_heads
    q = nn.DenseGeneral((self.attn_heads, head_dim))(q)
    k = nn.DenseGeneral((self.attn_heads, head_dim))(kv)
    v = nn.DenseGeneral((self.attn_heads, head_dim))(kv)
    return nn.dot_product_attention(q, k, v)

class AttnBlock(nn.Module):
  """Attention block with skip connection."""

  attn_type: str
  attn_heads: int = 4

  @nn.compact
  def __call__(self, h_in: jnp.ndarray):
    B, _, H, W, C = h_in.shape

    h = GroupNorm()(h_in)
    h0 = h[:, 0].reshape(B, H * W, C)
    h1 = h[:, 1].reshape(B, H * W, C)
    attn_layer = AttnLayer(attn_heads=self.attn_heads)

    if self.attn_type == 'self':
      h0 = attn_layer(q=h0, kv=h0)
      h1 = attn_layer(q=h1, kv=h1)
    elif self.attn_type == 'cross':
      h0 = attn_layer(q=h0, kv=h1)
      h1 = attn_layer(q=h1, kv=h0)
    else:
      raise NotImplementedError(self.attn_type)

    h = jnp.stack([h0, h1], axis=1)
```

```python
        h = h.reshape(B, 2, H, W, -1)
        h = nn.DenseGeneral(C, axis=(-2, -1), kernel_init=out_init_scale())(h)
        return (h + h_in) / onp.sqrt(2)

class XUNetBlock(nn.Module):
    """X-UNet block."""

    features: int
    use_attn: bool = False
    attn_heads: int = 4
    dropout: float = 0.0

    @nn.compact
    def __call__(self, x: jnp.ndarray, emb: jnp.ndarray, *, train: bool):
        h = ResnetBlock(features=self.features, dropout=self.dropout)(
            x, emb, train=train
        )

        if self.use_attn:
            h = AttnBlock(attn_type='self', attn_heads=self.attn_heads)(h)
            h = AttnBlock(attn_type='cross', attn_heads=self.attn_heads)(h)

        return h

class ConditioningProcessor(nn.Module):
    """Process conditioning inputs into embeddings."""

    emb_ch: int
    num_resolutions: int
    use_pos_emb: bool = True
    use_ref_pose_emb: bool = True

    @nn.compact
    def __call__(self, batch: dict[str, jnp.ndarray], cond_mask: jnp.ndarray):
        B, H, W, _ = batch['x'].shape

        # Log signal-to-noise-ratio embedding.
        logsnr = jnp.clip(batch['logsnr'], -20.0, 20.0)
        logsnr = 2.0 * jnp.arctan(jnp.exp(-logsnr / 2.0)) / onp.pi
        logsnr_emb = posenc_ddpm(logsnr, emb_ch=self.emb_ch, max_time=1.0)
        logsnr_emb = nn.Dense(self.emb_ch)(logsnr_emb)
        logsnr_emb = nn.Dense(self.emb_ch)(nonlinearity(logsnr_emb))

        # Pose embeddings.
        world_from_cam = v3d.Transform(R=batch['R'], t=batch['t'])
        cam_spec = v3d.PinholeCamera(resolution=(H, W), K=batch['K'])
        rays = v3d.Camera(spec=cam_spec, world_from_cam=world_from_cam).rays()

        pose_emb_pos = posenc_nerf(rays.pos, min_deg=0, max_deg=15)
        pose_emb_dir = posenc_nerf(rays.dir, min_deg=0, max_deg=8)
        pose_emb = jnp.concatenate([pose_emb_pos, pose_emb_dir], axis=-1)

        # Enable classifier-free guidance over poses.
        D = pose_emb.shape[-1]
        assert cond_mask.shape == (B,)
        cond_mask = cond_mask[:, None, None, None, None]
        pose_emb = jnp.where(cond_mask, pose_emb, jnp.zeros_like(pose_emb))
```

```python
    # Learnable position embeddings over (H, W) of frames (optional).
    if self.use_pos_emb:
      pos_emb = self.param(
          'pos_emb',
          nn.initializers.normal(stddev=1.0 / onp.sqrt(D)),
          (H, W, D),
          pose_emb.dtype,
      )
      pose_emb += pos_emb[None, None]

    # Binary embedding to let the model distinguish frames (optional).
    if self.use_ref_pose_emb:
      first_emb = self.param(
          'ref_pose_emb_first',
          nn.initializers.normal(stddev=1.0 / onp.sqrt(D)),
          (D,),
          pose_emb.dtype,
      )[None, None, None, None]

      other_emb = self.param(
          'ref_pose_emb_other',
          nn.initializers.normal(stddev=1.0 / onp.sqrt(D)),
          (D,),
          pose_emb.dtype,
      )[None, None, None, None]

      pose_emb += jnp.concatenate([first_emb, other_emb], axis=1)

    # Downsample ray embeddings for each UNet resolution.
    pose_embs = []
    for i_level in range(self.num_resolutions):
      pose_embs.append(
          nn.Conv(
              features=self.emb_ch,
              kernel_size=(1, 3, 3),
              strides=(1, 2**i_level, 2**i_level),
          )(pose_emb)
      )

    return logsnr_emb, pose_embs

class XUNet(nn.Module):
  """Our proposed XUNet architecture."""

  ch: int = 256
  ch_mult: tuple[int] = (1, 2, 2, 4)
  emb_ch: int = 1024
  num_res_blocks: int = 3
  attn_resolutions: tuple[int] = (8, 16, 32)
  attn_heads: int = 4
  dropout: float = 0.1
  use_pos_emb: bool = True
  use_ref_pose_emb: bool = True

  @nn.compact
  def __call__(
      self,
```

```python
    batch: dict[str, jnp.ndarray],
    *,
    cond_mask: jnp.ndarray,
    train: bool,
):
  _, _, _, C = batch['x'].shape
  num_resolutions = len(self.ch_mult)

  logsnr_emb, pose_embs = ConditioningProcessor(
      emb_ch=self.emb_ch,
      num_resolutions=num_resolutions,
      use_pos_emb=self.use_pos_emb,
      use_ref_pose_emb=self.use_ref_pose_emb,
  )(batch, cond_mask)
  del cond_mask

  h = jnp.stack([batch['x'], batch['z']], axis=1)
  h = nn.Conv(self.ch, kernel_size=(1, 3, 3), strides=(1, 1, 1))(h)

  # Downsampling.
  hs = [h]
  for i_level in self.ch_mult:
    emb = logsnr_emb[..., None, None, :] + pose_embs[i_level]

    for _ in range(self.num_res_blocks):
      use_attn = h.shape[2] in self.attn_resolutions
      h = XUNetBlock(
          features=self.ch * self.ch_mult[i_level],
          dropout=self.dropout,
          attn_heads=self.attn_heads,
          use_attn=use_attn,
      )(h, emb, train=train)
      hs.append(h)

    if i_level != num_resolutions - 1:
      emb = logsnr_emb[..., None, None, :] + pose_embs[i_level + 1]
      h = ResnetBlock(dropout=self.dropout, resample='down')(
          h, emb, train=train
      )
      hs.append(h)

  # Middle.
  emb = logsnr_emb[..., None, None, :] + pose_embs[-1]
  use_attn = h.shape[2] in self.attn_resolutions
  h = XUNetBlock(
      features=self.ch * self.ch_mult[i_level],
      dropout=self.dropout,
      attn_heads=self.attn_heads,
      use_attn=use_attn,
  )(h, emb, train=train)

  # Upsampling.
  for i_level in reversed(range(num_resolutions)):
    emb = logsnr_emb[..., None, None, :] + pose_embs[i_level]

    for _ in range(self.num_res_blocks + 1):
      use_attn = hs[-1].shape[2] in self.attn_resolutions
      h = jnp.concatenate([h, hs.pop()], axis=-1)
      h = XUNetBlock(
```

```
          features=self.ch * self.ch_mult[i_level],
          dropout=self.dropout,
          attn_heads=self.attn_heads,
          use_attn=use_attn,
      )(h, emb, train=train)

    if i_level != 0:
      emb = logsnr_emb[..., None, None, :] + pose_embs[i_level]
      h = ResnetBlock(dropout=self.dropout, resample='up')(
          h, emb, train=train
      )

  # End.
  assert not hs
  h = nonlinearity(GroupNorm()(h))
  return nn.Conv(
      C,
      kernel_size=(1, 3, 3),
      strides=(1, 1, 1),
      kernel_init=out_init_scale(),
  )(h)[:, 1]
```

## 7 HYPERPARAMETERS

We now detail hyperparameter choices across our experiments. These include choices specific to the neural architecture, the training procedure, and also choices only relevant during inference.

### 7.1 NEURAL ARCHITECTURE

For our neural architecture, our main experiments use `ch=256` (∼471M params), and we also experiment with `ch=448` (∼1.3B params) in Section 4. One of our early findings that we kept throughout all experiments in the paper is that `ch_mult=(1, 2, 2, 4)` (i.e., setting the lowest UNet resolution to 8x8) was sufficient to achieve good sample quality, wheras most prior work includes UNet resolutions up to 4x4. We swept over the rest of the hyperparameters with the input and target views downsampled from their original 128x128 resolution to the 32x32, selecting values that led to the best qualitative improvements. These values are the default values present in the code we provide for the `XUNet` module in Section 6. We generally find that the best hyperparameter choices at low-resolution experiments transfer well when applied to the higher resolutions, and thus recommend this strategy for cheaper and more practical hyperparameter tuning.

For the ∼1.3B parameter models, we tried increasing the number of parameters of our proposed UNet architecture through several different ways: increasing the number of blocks per resolution, the number of attention heads, the number of cross-attention layers per block, and the base number of hidden channels. Among all of these, we only found the last to provide noticeably better sample quality. We thus run 3D consistency scoring for models scaled this way, with channel sizes per UNet resolution of $448 \times [1, 2, 2, 4]$ instead of $256 \times [1, 2, 2, 4]$ (we could not fit `ch=512` in TPUv4 memory without model parallelism). On cars, we find that the smaller model reported in the rest of the paper is comparable in quality to the ∼1.3B parameter model, though on chairs, we do observe a significant improvement on 3D consistency, both visually and quantitatively (see Table 6).

### 7.2 TRAINING

Following Equation 3 in the paper, our neural network attempts to model the noise $\epsilon$ added to a real image in order to undo it given the noisy image. Other parameterizations are possible, e.g., predicting $x$ directly rather than predicting $\epsilon$, though we did not sweep over these choices. For our noise schedule, we use a cosine-shaped log signal to noise ratio that monotonically decreases from 20 to -20. It can be implemented in JAX as follows:

```python
def logsnr_schedule_cosine(t, *, logsnr_min=-20., logsnr_max=20.):
  b = onp.arctan(onp.exp(-.5 * logsnr_max))
  a = onp.arctan(onp.exp(-.5 * logsnr_min)) - b
  return -2. * jnp.log(jnp.tan(a * t + b))
```

Additionally:

- We use a learning rate with peak value 0.0001, using linear warmup for the first 10 million examples (where one batch has `batch_size` examples), following Karras et al. (2022).
- We use a global batch size of 128.
- We train each batch element as an *unconditional* example 10% of the time to enable classifier-free guidance. This is done by overriding the conditioning frame to be at the maximum noise level. We note other options are possible (e.g., zeroing-out the conditioning frame), but we chose the option which is most compatible with our neural architecture.
- We use the Adam optimizer (Kingma & Ba, 2014) with $\beta_1 = 0.9$ and $\beta_2 = 0.99$.
- We use EMA decay for the model parameters, with a half life of 500K examples (where one batch has `batch_size` examples) following Karras et al. (2022).

### 7.3 SAMPLING

While Ho et al. (2020) note that their ancestral sampler can be used with any variances between $\tilde{\beta}_t$ (variances of $q(z_s | z_t, x)$) and $\beta_t$ (variances of the underlying SDE), we simply use the former as those correspond to their ancestral sampler. We do not observe this choice to have a significant

effect sample quality. All our samples are generated with 256 denoising steps. As is standard in the literature, we also clip each predicted $x$ at each denoising step to the normalized range of the images $[-1, 1]$.

**Classifier-free guidance**. Our models use classifier-free guidance (Ho & Salimans, 2021), as we find that small guidance weights help encourage 3D consistency further. All our models were trained unconditionally with a probability of 10% for each minibatch element. For unconditional examples, we zero out the (positionally encoded) pose and replace the clean frame with standard Gaussian noise (leveraging our weight-sharing architecture, see Section 2.3). We swept over various guidance weights, and simply picked those where 3D inconsistency was qualitatively least apparent on sampled videos. For SRN cars, we use a guidance weight of 3.0, while for SRN chairs we use a weight of 2.0.

## 7.4 3D CONSISTENCY SCORING

Note that traditional NeRFs (Mildenhall et al., 2020) can be 3D inconsistent as the model allows for view-dependent radiance. We thus employ a simpler and faster to train version based on instant-NGP (Müller et al., 2022) without view dependent components, and with additional distortion and orientation loss terms for improved convergence (Barron et al., 2022; Verbin et al., 2022). Note that the specific implementation details of the neural field method chosen will affect the metrics considerably, so it is of utmost importance to apply the same method and hyperparameters for the neural fields to make 3D consistency scores comparable across models.

We design the neural fields as simple Multi-Layer Perceptrons (MLPs) of hidden size 64 and no skip connections. The density MLP has one hidden layer, while the MLP that predicts the color components uses 2 hidden layers. We use a learning rate of $0.01$ linearly decayed to $0.001$ for the first 100 steps. We use the Adam optimizer with weight decay set to 0.1 and clip gradients of norm exceeding 1.0. We only apply 1000 training steps for each scene and do not optimize camera poses. On each training step, we simply optimize over *all* available pixels, rather than sampling a random subset of pixels from all the training views.

To render the neural fields after training, we set near and far bounds to $t_n = \frac{3r_{\min}}{8}, t_f = \frac{3r_{\max}}{2}$, where $r_{\min}, r_{\max}$ are the minimum and maximum distances from the camera positions to the origin (center of each object) in the corresponding dataset. All renders post-training are also performed with differentiable volume rendering.

