# OpenReview forum: "Novel View Synthesis with Diffusion Models"
_ICLR.cc/2023/Conference — ICLR 2023 poster_

### Official Review · Reviewer_o2Dj · 2022-10-16

**Confidence:** 4
**Correctness:** 4
**Technical Novelty And Significance:** 3
**Empirical Novelty And Significance:** 3
**Recommendation:** 8

**Clarity, Quality, Novelty And Reproducibility:**

- The paper is easy to understand, well written and of high quality in general.
- It is also original in a sense that it reconceptualizes novel view synthesis as a diffusion process
- The authors provide detailed algorithmic information and code snippets for JAX in the supplemental materials, which should make reproduction easier

**Strength And Weaknesses:**

Strengths:
- A metric for qualitatively evaluating 3D consistency of novel view synthesis is a great contribution to the field.
- It is exciting to see that the proposed diffusion method is able to infer very consistent 3D structure only from view pairs.
- The cross attention scheme in the presented X-UNet makes sense in the given setting.
- The paper is very well written, well motivated and gives a useful overview about the area, structuring some related methods
- The generated novel views have impressive visual quality
- The individual contributions (stochastic sampling, diffusion and architecture changes) are evaluated in ablation studies.

Weaknesses:
- The authors present a novel metric for evaluating 3D consistency but put not enough effort on evaluating that metric itself. At least I would expect a comparison like in Table 2 which compares related methods under the new metric. Of course, it is expected that geometry-based methods outperform the presented method. It would be interesting to see how close 3DiM gets though. Also might be a good sanity check for the metric.
- The authors provide some details about the metric in the supplemental materials, in which they mention that they use Instant-NGP instead of NeRF but provide no further information about for example efficiency. I wonder how different architectures compare in a speed/metric quality tradeoff.
- Some details of the experimental setup are not clear to me (see below)
- There are very few qualitative results in the paper and supplemental. Only two object categories of one dataset are shown and compared.
- There is no dedicated related work section. Most important related work is cited and sufficiently introduced in the introduction but many works are missing.


**Summary Of The Paper:**

The paper presents a geometry-free method for novel view synthesis of objects utilizing probabilistic diffusion. Main contributions are a novel scheme for stochastic sampling of conditioning examples, a refined U-Net architecture for denoising, and a new metric for evaluating 3d consistency of novel view synthesis approaches.

**Summary Of The Review:**

In summary, I tend to vote for accepting the paper as it provides novel insight into geometry-free novel view synthesis (e.g. diffusion is able to capture 3D information based on 2D conditioning) and a list of smaller useful contributions, such as a novel metric for evaluating 3D consistency. It would be great if the authors could address the weaknesses, especially in experimental evaluations, as well as answer the following questions.

Questions and requests:
- I hope that the method actually starts from pure noise when generating novel views with a trained model (inference). The paper does not make that clear and Figure 2 and Figure 4 suggest that it receives a noisy version of the target. If that is the case it would heavily dampen significance.
- I would be interested in an analysis of cross attention pairs in the U-Net. The authors have the suspicion that it helps exploiting symmetries. A qualitative analysis of what-attends-to-what would increase insight.

---

> ### Author Response · Authors · 2022-11-17
> **Reply to reviewer o2Dj**
>
> Thank you for your valuable review. We address some of your comments below:
>
> > The authors present a novel metric for evaluating 3D consistency but put not enough effort on evaluating that metric itself. At least I would expect a comparison like in Table 2 which compares related methods under the new metric. Of course, it is expected that geometry-based methods outperform the presented method. It would be interesting to see how close 3DiM gets though. Also might be a good sanity check for the metric.
>
> The reason we didn’t run the baseline methods on our proposed evaluation mechanism is that we know they are 3D consistent by design due to the use of NeRF. For comparison purposes, the ground truth is probably the best choice to compare against (which we did), since the output views from prior work can be very blurry. Recall that FID here is comparing NeRF renders to model outputs, not ground truth (see the desiderata in the paper). In the paper, we discuss this limitation of the evaluation scheme: comparability is possible under the assumption that the views have reasonably low FID scores, i.e., the individual views resemble ground truth views when observed individually rather than as videos.
> We are actively trying to acquire outputs from prior geometry-free approaches where it would be prudent to run them through our proposed evaluation scheme.
>
>
> > The authors provide some details about the metric in the supplemental materials, in which they mention that they use Instant-NGP instead of NeRF but provide no further information about for example efficiency. I wonder how different architectures compare in a speed/metric quality tradeoff.
>
> We also just opted for a neural field with a training procedure that’s well known to be quick, as we have to train one for each set of views (one per asset, just to be pedantic). Since neural field training is just for quantifying 3D consistency, what matters most is having a single, consistent type of neural field so we can compare between different models. Training much slower neural fields doesn’t really add value and would be very expensive in terms of GPU hours, considering the datasets have at least 500 test objects each.
>
>
> > There are very few qualitative results in the paper and supplemental. Only two object categories of one dataset are shown and compared.
>
> We remark that the SRN benchmark is a different dataset to the results we present on the entirety of ShapeNet. The reason that only two categories are presented is that this dataset only has cars and chairs (not other ShapeNet classes), and it is the dataset that lets us compare with the most prior works. Nevertheless, as we mentioned in our general comment to all reviewers, we are including many more samples for all of our models, as we agree we could’ve presented more. We also include new results on images from the internet where we do not have access to poses at test time, definitely going beyond cars and chairs.
>
>
> > I hope that the method actually starts from pure noise when generating novel views with a trained model (inference). The paper does not make that clear and Figure 2 and Figure 4 suggest that it receives a noisy version of the target. If that is the case it would heavily dampen significance.
>
> The model indeed starts from pure noise at generation time (see Figure 3). Figure 2 illustrates training, and Figure 3 illustrates inference. We have clarified these figures further in the paper and figure captions, noting that multiple reviewers had comments about this.

---

> > ### Comment · Reviewer_o2Dj · 2022-11-23
> > **Thanks**
> >
> > I thank the authors for their answers.
> >
> > I am not fully convinced by the reasoning why the metric was not evaluated on other methods. I think the paper would profit from it, especially if it is meant to be picked up by future work. For that, it would be very useful to know how it evaluates on these baselines. It would also serve as a sanity check, i.e. if it behaves as expected. Also, NeRF-based representation are not always 3D consistent/view consistent.  Some NeRF-style methods generalize much better to novel views than others/need less training views to become view consistent.
> >
> > This is mostly unrelated to the main contribution. My score and assessment stays unchanged.

---

### Official Review · Reviewer_wGc3 · 2022-10-24

**Confidence:** 4
**Correctness:** 3
**Technical Novelty And Significance:** 3
**Empirical Novelty And Significance:** 3
**Recommendation:** 6

**Clarity, Quality, Novelty And Reproducibility:**

The quality and clarity of the work are high. The problem and constraints are described clearly. The contrasts with other approaches or other works are also explained in depth. The choices made are each explained.

The originality of the work is moderately high. The original contributions are the choice of problem and model class (diffusion models), the modified U-Net, stochastic conditioning, and a novel metric for generation quality. Each individually is not a large contribution but taken together this work represents a significant exploration of the area.

The reproducibility of the work is sufficient and hyperparameters are provided in the supplementary work.

**Strength And Weaknesses:**

Strengths:
- The model results are very promising. The rendered views are detailed and geometrically consistent with the input views.
- The approach, details, and results are described clearly, including author reasoning for the choices they have made.

Weaknesses:
- The authors note that although one might expect to condition jointly on all available views, "We found this solution to perform poorly ... We also find that, as we increase the maximum number of input frames k, the worse the sample quality becomes." This is counterintuitive. It seems likely that rather than being a fundamental aspect of the problem, this is a weakness of the model architecture or something else. The value of one of the core contributions – stochastic conditioning – relies on this observation, so it would be better if the authors spend more time to experiment with and explain why this is the case, rather than taking it for granted.
- It seems like stochastic conditioning relies on having the number of diffusion steps be significantly greater than the number of input views. Additionally, it seems like the order in which the input views are used may have a significant impact on the resulting image and perhaps the resulting image quality or consistency. It would help if the authors investigate the impact of number of diffusion steps as well as impact of view ordering for stochastic conditioning.
- Between the two weaknesses listed previously, it seems like stochastic conditioning could prove to be a poor choice and have limited significance. If you reduce the number of diffusion steps (as many contemporary works aim to do), stochastic conditioning could prove to perform poorly, or be incredibly sensitive to the order of views. Similarly, if stochastic conditioning is simply a workaround for a network architecture that cannot scale to multiple input views, stochastic conditioning will quickly become irrelevant once the network architecture is improved.
- Relatively minor – The modified U-Net has several modifications; however, the ablations only compare *all* modifications against *no* modifications, without individually showing that the modifications are useful.

**Summary Of The Paper:**

The authors propose 3DiM, a diffusion model for novel view synthesis. The input to the model is a set of images with poses representing different views of the same object plus a target pose; the output of the model is a new image of the same object from the target pose. By applying the model repeatedly, the authors can generate many novel views.

The model consists of a U-Net-style diffusion image-to-image model. Notably, it only takes a single conditioning image as input, and in order to condition on multiple frames, the conditioning is varied through the diffusion process. The authors additionally modify the traditional U-Net used in diffusion modeling with a few architecture tweaks and show via an ablation that their modifications are an improvement over the default network structure.

Finally, the authors propose a new metric for measuring the 3D consistency of their model. This metric is measured by training a NeRF on a subset of the model outputs and evaluating by comparing the NeRF outputs to the rest of the model outputs.

**Summary Of The Review:**

Overall, the paper is above the acceptance threshold. It is well-written and reasoned, with a novel approach and interesting results which suggest a number of further research directions. However, as mentioned in the weaknesses section, one of the fundamental contributions of this work (stochastic conditioning) is underexplored; the need for it could be better justified and its behavior could be better evaluated and quantified. Given that this is fundamental to the approach and may significantly limit the significance of the work, I believe the paper is not *strongly* above the acceptance threshold.

---

> ### Author Response · Authors · 2022-11-17
> **Reply to reviewer wGc3**
>
> Thank you for your valuable review. We address some of the discussion regarding stochastic conditioning below.
>
> We agree that relying on many denoising steps is a limitation of stochastic conditioning and have discussed it more in the paper. We also consider that finding a better architecture to condition on and/or generate multiple views is important future work. Advancing the architecture was one of the key ingredients for our results to achieve significant quality, and we believe that advancing it more will cause further improvement if done properly. We still believe that stochastic conditioning will remain useful, though, due to various reasons:
> - Architectures that either process or generate more views entail more running time. Image-to-image training is itself useful (and can be sufficient, as we show) for its computational efficiency. It allows training at larger batch sizes and prevents the need for model parallelism. It also unlocks the ability to use cross-attention over feature maps, rather than attention over time, as we discuss in the paper.
> - Stochastic conditioning unlocks the ability to train from examples where only two views are available in the data (a concrete example could be applying 3DiM to stereo pairs, which are much more abundant than 3D data).
> - Stochastically conditioning on subsets of the previous views could help get across the practical limits of attention mechanisms (used in UNet), which is quadratic in sequence length and is well-known to degrade for very long sequences anyway (Gu et al., 2022).
> - There is new work (ERNIE, Feng et al., 2022) that appeared after submission time that utilizes similar techniques, interleaving different denoising experts. This is extremely similar in spirit to stochastic conditioning, applied to a different setting, so it is one datapoint showing that these techniques are a valuable contribution to the literature of diffusion models.
>
> > Additionally, it seems like the order in which the input views are used may have a significant impact on the resulting image and perhaps the resulting image quality or consistency.
>
> Regarding the order of generation, our results do not sort the frames by pose nearness. We still conducted this experiment after the submission deadline and didn’t observe much difference in quality between generating in order v.s. at random order. We don’t think this is too surprising as our image-to-image training is with pairs at randomly uniform camera positions around the object. This means that, despite that when the delta in pose is small the task should be easier, more often than not the model sees training examples where this delta is not so small.
>
> In general, we did not find that other additions to the stochastic conditioning pipeline to be of much significance either, e.g., making the conditioning frame more likely, introducing temperature to make nearby frames more likely, etc. qualitatively did not make a big difference. The crucial ingredient is to simply allow the model to visit various past frames. We thus opted to present and ablate the simplest version of stochastic conditioning– just sampling one of the previous frames with uniform probability.
>
> If the paper gets accepted, we will be sure to include some comparisons between different inference-time hyperparameters like these in the camera-ready version!

---

### Official Review · Reviewer_Am8T · 2022-10-24

**Confidence:** 3
**Correctness:** 3
**Technical Novelty And Significance:** 4
**Empirical Novelty And Significance:** 4
**Recommendation:** 6

**Clarity, Quality, Novelty And Reproducibility:**

The paper is mostly clear. The methodology is sound and principled. All three proposals (e.g. the stochastic conditioning, the X-UNet, and the evaluation protocol) seem to be original. Authors add code in the supplementary material.

**Strength And Weaknesses:**

Main Weaknesses:
- Figs. 1, 2, 4 and 5 are not referred to in the text. Partially due to this fact, it is a little hard to follow the overall functioning of the methodology. Initially, Fig. 2 seems to show that, yet the "green" block appears "inside" Fig. 3 diagrams. In the latter, the little "dice" is not mentioned in the caption and its meaning is not completely clear. It is also not clear how the X-UNet architecture (Fig. 4), relates to the other diagrams in Fig. 2 and 3. It could be a good idea to who the overall architecture initially and clearly relate its block with the following diagrams in the subsequent figures.
- The provided "webpage" in the supplementary material is not working properly. I also believe the supplementary should be in a separate pdf other than along with the main paper.


Main Strengths:
- The paper is well-written and based on very recent and updated literature.
- The proposed method is sound and tackles a relevant problem with an original method.
- The authors overcome the limitations of the standard evaluation metrics with the proposal o a new evaluation protocol.


**Summary Of The Paper:**

The paper explores the use of diffusion models for novel view synthesis. The authors introduce a geometry-free image-to-image model, dubbed X-UNet, based on a new stochastic conditioning sampling algorithm. They compare their work with state-of-the-art baselines on SRN ShapeNet dataset, apparently, achieving better qualitative results. Regarding quantitative scores, the authors point out some limitations of standardised metrics and, consequently, propose a new evaluation protocol.

**Summary Of The Review:**

The paper introduces a novel and well-principled method for novel view synthesis. The diagrams and overall architecture pipeline could be better presented. Qualitative results are better than state-of-the-art. However, quantitative scores fall short. Regarding this issue, the authors point out the reasons for this problem and propose a novel evaluation protocol. In summary, I believe the paper is worthy of publication conditioned to the improvement of presentation.

---

> ### Author Response · Authors · 2022-11-17
> **Reply to reviewer Am8T**
>
> Thank you for your valuable review. We address some of your comments below:
>
> > Figs. 1, 2, 4 and 5 are not referred to in the text. Partially due to this fact, it is a little hard to follow the overall functioning of the methodology. Initially, Fig. 2 seems to show that, yet the "green" block appears "inside" Fig. 3 diagrams. In the latter, the little "dice" is not mentioned in the caption and its meaning is not completely clear. It is also not clear how the X-UNet architecture (Fig. 4), relates to the other diagrams in Fig. 2 and 3. It could be a good idea to who the overall architecture initially and clearly relate its block with the following diagrams in the subsequent figures.
>
> We have updated the paper to have pointers between text and figures, and the figure captions themselves should be more clear now. We also clarify here, just to be explicit: the dice are simply used to represent the stochastic nature of selecting a conditioning view, and the green block represents the neural network. X-UNet is the green block itself. The reason why Figure 3 has multiple green blocks is that, at generation time, diffusion models run multiple forward passes to iteratively turn noise into an image (unlike in training, Figure 2, where diffusion models just learn to undo noise at random levels).
>
>
> > The provided "webpage" in the supplementary material is not working properly. I also believe the supplementary should be in a separate pdf other than along with the main paper.
>
> We have made fixes to the supplementary website, and it should be much more friendly to use now, especially in mobile mode. Please let us know if anything specific remains broken so we can fix it.
>
>
> We would greatly appreciate you letting us know if, after these fixes, there is more room for improvement regarding clarity and presentation, which seem to be the main concerns of this reviewer. We will engage with any comments as quickly as possible.

---

### Official Review · Reviewer_Zv2q · 2022-10-31

**Confidence:** 4
**Correctness:** 3
**Technical Novelty And Significance:** 2
**Empirical Novelty And Significance:** 2
**Recommendation:** 5

**Clarity, Quality, Novelty And Reproducibility:**

The paper includes supplementary material that explains the detail of the implementation. The proposed approach is reproducible.

**Details Of Ethics Concerns:**

At the time of ICLR submission, the project page of this paper was unveiled in the public domain with full authorship for a few days. @AC: Would this violate the ICLR submission policy?

**Strength And Weaknesses:**

**Strengths**
1. The paper introduces a new approach for the novel view synthesis using the diffusion model. The generated images are plausible and consistent over the various viewpoints. The approach shows compelling results on the ShapeNet dataset.
2. The approach introduces an interesting approach for rendering novel views from a single image. The paper explains the proposed idea in detail.
3. The authors proposed a new approach to measure view consistency. By applying NeRF with the viewing direction, the better reconstruction of the radiance field indicates that the input images are more consistent across viewing directions, which is a reasonable and good attempt.

**Weakness**
1. The paper exposition requires significant improvement. The paper does not read well. For instance, the caption Figure 3 states, "There are two main components to our sampling procedure 1) the autoregressive generation of multiple frames, and 2) the denoising process to generate each frame". However, it is unclear which step indicates which section in the main paper, and there is no description of how the two steps can be utilized for the training. It is also confusing what the subfigures for Step 1 and Step 2 indicate. I think step 2 explains Eq 5 and Eq 6 since the input of the denoising model is the random viewpoint of the clean image, and I think Sec 2.1 corresponds to Figure 2. Then, what is Step 1 indicate? What are dice in the figure mean? Why does the pose of the dices varies while the poses of the images next to the dices are not changing?
2. In addition, the paper requires an overview figure that can clearly show the flow of the training and testing phases. The figure should be consistent with the sentence in the abstract, "a pose conditional image-to-image diffusion model, which takes a source view and its pose as inputs, and generates a novel view for a target pose as output". The reader would expect the pipeline that takes the input image with its pose and target pose as the condition and the network to produce the novel view. The current figures need to explain how the trained network can be used to generate target view images. Instead, Figures 2 and 3 show additional clean images required for the training, and in Figure 4, the network does not require camera poses, but the network only requires noise and clean image pairs.
3. Please build a connection between sections and figures. For instance, I think \epsilon_\theta in Sec 2.1 is described to be used in Eq. 6 in Sec 2.2, right? It would be good to mention how the notations and equations can be connected, resulting in the final equation.
4. The limited dataset demonstrates the proposed approach, where the scenes are synthetic. Can the proposed method be applied to real images? For instance, I wonder if a bus on the road identified by a semantic segmentation mask can be view-interpolated. This would be tricky for the proposed approach since the synthetic images used for the training is biased toward specific camera intrinsic parameters. Given the image of the different camera intrinsic, not observed in the training scene, the view-interpolation would be failed. Another dataset that can be applied to the proposed approach is the DTU MVS dataset [Jensen et al. 2014]. The such dataset would be helpful to show the benefit of the proposed approach.
5. Another limitation of the proposed approach is that approach is trained with the view-aligned 3D shapes. The input image and its viewing direction need to be an input of the proposed framework, limiting the applicability of the proposed approach. How can the approach be applied when the initial pose of the object is not provided?
6. The technical contribution is not significant. It is an adaptation of the well-developed diffusion model [Salimans & Ho 2022, Ho et al. 2020, Kingma & Welling 2013] for the view of consistent image synthesis. The key modification is to provide random viewpoint images for denoising.
7. At the time of ICLR submission, the project page of this paper was unveiled in the public domain for a few days. @AC: Would this violate the ICLR submission policy?
8. I think the term "3D consistent" is misleading since the proposed approach is not utilizing 3D shapes. Instead, view consistency would be better terminology.
9. The approach requires 256 denoising steps to generate a single image. It would take significant inference time. The runtime of the proposed approach should be compared with the baseline approaches. The current version of the paper lacks computational comparison.
10. The proposed approach is actually performing worse than other baseline approaches, such as PixelNeRF and CodeNeRF, in terms of PSNR and SSIM. Although PixelNeRF performs better in terms of PSNR, the result in Figure 5 is much worse than the images shown in the PixelNeRF. It is suspicious the results are cherry-picked images. Authors claim that it is not necessary to achieve the top PSNR, but this is not very convincing since the view interpolation has the ground truth solution. The comparison with the GT solution is essential.
11. Minor comment: CodeNeRF is not mentioned on page 6.
12. In Figure 6, it is questionable that the black-silhouette images can be interpolated as the orange-colored chairs. Is this the ablation study tested with unseen images? I suspect the approach is overfitted to the training dataset.

**Reference**

[Jensen et al. 2014] Rasmus Jensen, Anders Dahl, George Vogiatzis, Engil Tola, and Henrik Aanæs. Large scale multi-view stereopsis evaluation. In CVPR, pages 406–413, 2014.



**Summary Of The Paper:**

This paper introduces an approach for rendering novel views from a single image. The approach is based on a pose-conditional image-to-image diffusion model. The paper shows the proposed approach can be generalized to the shapes that are not seen during the test time. The results of the proposed approach are superior then the results optained by prior art.

**Summary Of The Review:**

The paper introduced an interesting approach for the view synthesis of an object using the diffusion model. It is also interesting to see that the additional random view images would help to generate view-consistent images. However, the paper requires significant revision in the exposition of the proposed approach to be more clearly to be understood.

---

> ### Author Response · Authors · 2022-11-17
> **Reply to reviewer Zv2q (part 1 / 2)**
>
> Thank you for your valuable review. We have modified the paper to incorporate a lot of the feedback from the reviewers regarding presentation and clarity. Please find below our response to your questions and concerns.
>
>
> > The paper does not read well. For instance, the caption Figure 3 states, "There are two main components to our sampling procedure 1) the autoregressive generation of multiple frames, and 2) the denoising process to generate each frame". However, it is unclear which step indicates which section in the main paper, and there is no description of how the two steps can be utilized for the training.
>
> Neither the iterative denoising process nor the autoregressive generation of frames are part of training.  We have now clarified that Figure 3 exclusively illustrates inference, and have further clarified that “Step 1”, “Step 2”, etc. (represented vertically in the figure) illustrate the autoregressive component– they are not equations (5) and (6), which *together* illustrate a single denoising step (represented horizontally in the diagram). The dice in the diagram illustrate that the conditioning view is drawn randomly. The caption also explicitly states that the poses are omitted from the figure– this was a deliberate choice to avoid a confusing diagram with too many moving parts. We have also added cross-references between figures and the paper now. We hope this helps with clarity, and are grateful for any other questions or feedback to ensure our work is easier to read.
>
>
> > In addition, the paper requires an overview figure that can clearly show the flow of the training and testing phases.
>
> Figure 2 illustrates training, and Figure 3 illustrates inference. We believe that merging them into one figure would be too cluttered and difficult to digest for the reader. We appreciate your comment about the abstract setting a wrong expectation about the model’s inputs and outputs, and have clarified that the image-to-image setup corresponds to training. Please see our response to the immediately previous point, where we clarify the figures themselves.
>
>
> > Please build a connection between sections and figures.
>
> Done (see the first point).
>
>
> > Can the proposed method be applied to real images?
>
> As we discuss in the conclusion, non-synthetic datasets (especially those at large scale) have estimated poses that tend to be very noisy. There is an additional obstacle to consider: it can be difficult to obtain datasets with a sufficiently large number of assets to train a generative model (e.g., the dataset the reviewer pointed to has less than 150 scenes). We thus believe tackling this problem to be worthy of an entire research project of its own. We still agree that the constant intrinsics are a limitation, and have added more discussion about those and other specific limitations of 3DiM regarding the cameras it can and cannot handle. We note this is expected though, as 3DiM is geometry-free and is limited to handle pose distributions it sees during training.
>
>
> > How can the approach be applied when the initial pose of the object is not provided?
>
> Sharp, high-quality 3D generation from a single view (with poses) was already extremely challenging for prior work, so we believe that even with poses, the contribution of 3DiM is significant. Regardless, we have engaged with the reviewer’s request. While we do not have a solution to train without poses, we have included new results where 3DiMs are trained with relative poses and test it on images without known poses (as the input to the model now is only the delta in pose we wish to change by, as opposed to both a source and target pose). To demonstrate this, we show results from images we found on the internet, where we have no way of knowing pose. We realize that 3DiM samples are far from perfect in this out-of-distribution setting compared to the rest of our results, but still hope that the reviewer can appreciate both the potential for our approach in this setting and our engagement with this request which is beyond the original scope of the paper.
>
>
> > At the time of ICLR submission, the project page of this paper was unveiled in the public domain for a few days.
>
> ICLR does not have a social media ban like CVPR and other conferences, and does not discourage authors from publicly discussing their work. Like many other ICLR submissions, the project page actually remains public and de-anonymized. Please do NOT look it up in order to maintain the spirit of double-blind reviews. None of the materials disclosed to the reviewers reveal who we are.

---

> > ### Author Response · Authors · 2022-11-17
> > **Reply to reviewer Zv2q (part 2 / 2)**
> >
> > > I think the term "3D consistent" is misleading since the proposed approach is not utilizing 3D shapes. Instead, view consistency would be better terminology.
> >
> > “View consistency” might read as ambiguous? It could refer to alignment to the conditioning image, or to other generated views. “3D consistency” entails both, and is standard in the literature (e.g., it is common to say that NeRF is 3D consistent). We take additional care to say “approximate 3D consistency” wherever we refer to 3DiM; we do not wish to make any misleading claims, and have corrected a few slips where we did not specify that 3D consistency is only approximate.
> >
> >
> > > The approach requires 256 denoising steps to generate a single image. It would take significant inference time.
> >
> > We agree there is a lot of room for future work on making 3DiM faster, and that stochastic conditioning might prevent step reduction methods from being too successful. We have discussed this limitation further in the paper and in comments to other reviewers (e.g., please see our response to reviewer wGc3).
> >
> >
> > > The proposed approach is actually performing worse than other baseline approaches, such as PixelNeRF and CodeNeRF, in terms of PSNR and SSIM. [...] It is suspicious the results are cherry-picked images. Authors claim that it is not necessary to achieve the top PSNR, but this is not very convincing since the view interpolation has the ground truth solution. The comparison with the GT solution is essential.
> >
> > Please see our main comment to all reviewers; we have included more samples to hopefully convince reviewers that outputs from 3DiM are indeed sharp and well-aligned to the given inputs.
> > The reviewer additionally states that achieving the top PSNR is essential. However: if I give you the bottom of a chair, how do you know the color of the seat, or if it has a pillow that is occluded? If I give you the front of a car, how do you know if the lights at the back are on or off, or if there’s a hanging wheel? If I give you any object that looks small, is it truly small or just far away?
> > This is our main argument in the paper: the conditional distribution of views given a single view can be very multimodal, i.e., multiple outputs consistent with the input are plausible. Even a generative model that perfectly models the data distribution will not match the ground truth and top PSNR or any other reconstruction-based metric. If the reviewer has any more specific reasons to believe otherwise, we are happy to discuss further.
> >
> >
> > > Minor comment: CodeNeRF is not mentioned on page 6.
> >
> > Done. Thank you for pointing this out.
> >
> >
> > > In Figure 6, it is questionable that the black-silhouette images can be interpolated as the orange-colored chairs. Is this the ablation study tested with unseen images? I suspect the approach is overfitted to the training dataset.
> >
> > All our results are on unseen objects (i.e., absent in the training data), including every ablation. This is what “novel view synthesis” means.

---

> ### Comment · Reviewer_Zv2q · 2022-12-12
> **I have read the rebuttal**
>
> I acknowledge the author's rebuttal. The authors improved the paper's exposition after the rebuttal phase. However, still many concerns are not fully addressed.
>
> Authors claim that qualitative assessment is an important measure (sharp and clean images), but my concern is that even though the reconstructed images by the proposed approach are sharp and clean, PSNR gets worse than PixelNeRF, which generates blurred images.
> Authors said:
> > However: if I give you the bottom of a chair, how do you know the color of the seat, or if it has a pillow that is occluded? If I give you the front of a car, how do you know if the lights at the back are on or off, or if there's a hanging wheel? If I give you any object that looks small, is it truly small or just far away?
>
> Then, why can PixelNeRF still generate more reasonable images in terms of PSNR and SSIM than the results of the proposed approach? Authors claim that PixelNeRF generates blurred images. If so, the better PSNR and SSIM of PixelNeRF should indicate better color/geometry-consistent images, because it is the only factor to achieve better PSNR and SSIM if the images get blurred. In this sense, the generated images by the proposed approach may be considered as well-hallucinated sharp images, so evaluation based on the quality assessment should be done carefully.
>
> More importantly, the technical contribution is not significant. It is an adaptation of the well-developed diffusion model [Salimans & Ho 2022, Ho et al. 2020, Kingma & Welling 2013] for the view of consistent image synthesis. The key modification is to provide random viewpoint images for denoising.
>
> In addition, the approaches are not demonstrated with real-world images. Authors said
> > it can be difficult to obtain datasets with a sufficiently large number of assets to train a generative model
>
> But it is not convincing. I wonder if the authors actually try to find a large-scale object dataset like CO3D (https://ai.facebook.com/datasets/CO3D-dataset/), which is large enough for the generative models. The authors said, "estimated poses that tend to be very noisy". This may reveal that the proposed approach is very sensitive to the camera poses and why the demonstration is mainly conducted with synthetically rendered images.
>
> Overall, I still have the same level of concern about the issue of the paper. My score remains unchanged.

---

### Author Response · Authors · 2022-11-17
**Reply to all reviewers**

Dear reviewers,

Thank you for your valuable feedback. We’ll engage with each review separately, but will discuss common questions and concerns here.

First we include a summary of the changes:
- We have included new results on images with unknown pose at test time. To demonstrate this, we test 3DiM on real photorealistic images with white backgrounds we found on the internet, rather than using ShapeNet images to test the model. We trained a 471M parameter 3DiM on all of ShapeNet that only takes relative poses, rather than source and target absolute poses. We rendered 128 view pairs for each object in the training data, at random orientations, scales, and throwing in hue augmentation to make the model as robust as possible to this out-of-distribution setting. All renders are with 256 denoising steps as usual. Results are included in the supplementary website, including failure modes.
- We noted some concern about us not presenting sufficient samples from our models. We have now included many more samples in the supplementary website for the models presented in the paper (SRN cars, SRN chairs, and the model we trained on the entirety of ShapeNet). We made sure to include failure modes as well.
- We additionally noted that multiple reviewers were confused about our figures and expressed that some of the figures did not refer to the paper and vice versa. This is now fixed, and figures have been clarified further.
We have added more discussion regarding the limitations of 3DiM in the conclusion of the paper. Namely, multiple reviewers pointed out that stochastic conditioning requires many denoising steps, and that due to the geometry-free setup of 3DiM, it is difficult to handle cameras unseen during training (e.g., varying focal lengths, sensor width, object scales, etc.)
- The website itself should be usable in mobile devices now, and we provide a fully anonymized URL so it’s much more convenient for the reviewers to navigate:
https://anonymous-3dim-authors.github.io/

We realize our samples from out-of-distribution images (and hence without known poses), as requested by reviewer Zv2q, are very far from perfect. One main challenge is that it is difficult to find synthetic 3D datasets that have more objects and realism than ShapeNet, and we likely need more data to improve model robustness in this out-of-distribution setting. Still, we hope that our results demonstrate the potential of 3DiM in more challenging contexts such as this one (especially if trained on a dataset beyond ShapeNet on realism and/or scale) and that just being able to do 360 spins for images in the wild of this quality and diversity (with a single model) is unprecedented to the best of our knowledge.

---

### Decision · Program_Chairs · 2023-01-20

**Decision:**

Accept: poster

**Justification For Why Not Higher Score:**

The paper is promising but it needs more work and evaluation.

**Justification For Why Not Lower Score:**

The proposed method for view synthesis using diffusion models and the evaluation metrics proposed in the paper will benefit the ICLR community.

**Metareview: Summary, Strengths And Weaknesses:**

The paper proposes the use of diffusion models to synthesize novel views from 2D images. The training is of a score function on pairs of views and their positions where the goal of the training is to denoise a view conditioned on another clean view. The main contributions of the work are: 1) stochastic conditioning sampler on multiple views in random order 2) XUnet for the score network 3)  3D consistency scoring that measures the 3D consistency of a generated object by  applying NeRF with the viewing direction, the better reconstruction of the radiance field indicates that the input images are more consistent across viewing directions.

The proposed solution in the paper for generating consistent views opens the door for new applications of new views synthesis from single image.

Reviewers agreed that the main contributions of the paper are moderately innovative and not well studied in the paper: the stochastic conditioning needs a lot of steps  and diffusion steps to generate a view, and can be very sensitive to the conditioning. The evaluation is done on synthetic objects and views and the metric proposed was only only used to evaluate the proposed method and not other baselines.

Overall the positive overweight the negative but  the paper  will benefit from 1)  adding ablation on the hyperparameters of the synthesis and in 2)  its evaluation of other baselines with the proposed metrics 3 ) evaluation on real world datasets . Weak Accept.

**Note From Pc:**

if the above contains the word "oral" or "spotlight" please see: "oral" presentation means -> notable-top-5% and "spotlight" means -> notable-top-25%. As stated in our emails, we are disassociating presentation type from AC recommendations

**Summary Of Ac-Reviewer Meeting:**

N/A